# Crosstalk between Renal and Vascular Calcium Signaling: The Link between Nephrolithiasis and Vascular Calcification

**DOI:** 10.3390/ijms22073590

**Published:** 2021-03-30

**Authors:** Chan-Jung Liu, Chia-Wei Cheng, Yau-Sheng Tsai, Ho-Shiang Huang

**Affiliations:** 1Department of Urology, National Cheng Kung University Hospital, College of Medicine, National Cheng Kung University, Tainan 704302, Taiwan; dragon2043@hotmail.com (C.-J.L.); imjohnny7639@gmail.com (C.-W.C.); 2Institute of Clinical Medicine, College of Medicine, National Cheng Kung University, Tainan 704302, Taiwan; yaustsai@mail.ncku.edu.tw; 3Center for Clinical Medicine Research, National Cheng Kung University Hospital, Tainan 704302, Taiwan

**Keywords:** calcium signal, calcium homeostasis, kidney stone, nephrolithiasis, vascular calcification, calcium-sensing receptor, hypercalciuria, osteo-/chondrogenic transdifferentiation

## Abstract

Calcium (Ca^2+^) is an important mediator of multicellular homeostasis and is involved in several diseases. The interplay among the kidney, bone, intestine, and parathyroid gland in Ca^2+^ homeostasis is strictly modulated by numerous hormones and signaling pathways. The calcium-sensing receptor (CaSR) is a G protein–coupled receptor, that is expressed in calcitropic tissues such as the parathyroid gland and the kidney, plays a pivotal role in Ca^2+^ regulation. CaSR is important for renal Ca^2+^, as a mutation in this receptor leads to hypercalciuria and calcium nephrolithiasis. In addition, CaSR is also widely expressed in the vascular system, including vascular endothelial cells (VECs) and vascular smooth muscle cells (VSMCs) and participates in the process of vascular calcification. Aberrant Ca^2+^ sensing by the kidney and VSMCs, owing to altered CaSR expression or function, is associated with the formation of nephrolithiasis and vascular calcification. Based on emerging epidemiological evidence, patients with nephrolithiasis have a higher risk of vascular calcification, but the exact mechanism linking the two conditions is unclear. However, a dysregulation in Ca^2+^ homeostasis and dysfunction in CaSR might be the connection between the two. This review summarizes renal calcium handling and calcium signaling in the vascular system, with a special focus on the link between nephrolithiasis and vascular calcification.

## 1. Introduction

Calcium (Ca^2+^) is the most abundant mineral in the human body. As a result, it participates in several physiological and pathological processes. More than 99% of Ca^2+^ in the entire body and 85% of PO4^3−^ are combined as hydroxyapatite and deposited in bone under tight regulation [1]. The remaining 1% of Ca^2+^ can be found in the blood, the extracellular fluid (ECF), and soft tissues. Serum Ca^2+^ concentration is strictly maintained in the range between 8.5 and 10.5 mg/dL under the controls of parathyroid hormone (PTH), calcitonin, and 1,25-dihydroxyvitamin D (1,25(OH)_2_D_3_) [2]. Furthermore, serum levels of PTH and 1,25(OH)_2_D_3_ are regulated by the calcium-sensing receptor (CaSR), located in the parathyroid gland and kidney. CaSR can specifically detect a slight increase in serum Ca^2+^ and it can be activated by Ca^2+^ with low affinity [3]. Following CaSR activation, PTH secretion and renal Ca^2+^ reabsorption will significantly decrease. Decreased PTH level directly results in decreased Ca^2+^ reabsorption by the kidney, decreased Ca^2+^ resorption from bone, and decreased Ca^2+^ absorption in the intestine. As a result, this complex and precise modulatory mechanism is the interplay among the kidney, bone, intestine, and parathyroid gland. Ca^2+^ homeostasis is associated with multicellular dysfunctions, including cellular apoptosis, mitochondrial dysfunction, and oxidative stress [4,5,6]. Furthermore, a dysregulation of Ca^2+^ may also contribute to the pathogenesis of numerous diseases, including malignancy, diabetes mellitus (DM), and hypertension (HTN) [7,8,9]. Pathological mineralization is another consequence of Ca^2+^ dysregulation that can occur in vessels, kidneys, and many other organs. Vascular calcification is associated with a high cardiovascular-related morbidity and mortality, and can be a consequence of aging, chronic kidney disease (CKD), and HTN [10]. The pathophysiology of vascular calcification might be mainly related to osteo-/chondrogenic differentiation, which is partially modulated by the CaSR. Calcification in the kidney can result in nephrolithiasis (the presence of solid stones in the collecting system of the kidney) and nephrocalcinosis (the deposition of Ca^2+^ in the kidney parenchyma). The most common composition of renal stone is calcium oxalate (CaOx), followed by calcium phosphate (CaP). Although the exact mechanism of kidney stone formation is still unclear, Randall’s plaque theory is one of the most accepted hypotheses [11]. Randall’s plaque is an exposed CaP lesion without coated urothelium on the renal papilla and is frequently observed among stone formers [12]. These plaques are thought to serve as a nidus for CaOx overgrowth and urinary stone formation. The process of stone formation is frequently related to hypercalciuria, which is associated with renal Ca^2+^ handling and the renal CaSR [13]. In recent years, emerging epidemiological studies have disclosed the strong link between vascular calcification and nephrolithiasis; however, the exact mechanism is still largely unclear [14]. As the CaSR plays a pivot role in the regulation of calcification in various organs, a dysregulation in Ca^2+^ signaling could be considered as the connection between vascular calcification and nephrolithiasis. Thus, we have conducted this narrative review, which is based on the articles retrieved from the databases Medline and PubMed using the search terms “calcium signaling”, “calcium sensing receptor”, “vascular calcification”, “kidney stone” and “nephrolithiasis” for the topic.

In this review, we first briefly review the Ca^2+^ signaling pathway in the kidney and the role of renal CaSR. Owing to the critical role of CaSR in controlling physiological and pathological calcifications, we also addressed the impact of Ca signaling on vascular calcification and nephrolithiasis. Finally, we summarized the extensive literature on the link between vascular calcification and nephrolithiasis, and reviewed the possible hypothesis that may explain this link and the role played by the CaSR in this connection.

## 2. The Physiological Pathway of Ca^2+^ Signaling in Kidney

Owing to the high expression of the CaSR, the kidney is considered to be a calcium-sensing organ that can monitor the urine and serum levels of Ca^2+^ [15]. Elevated serum Ca^2+^ will directly lead to increased urinary Ca^2+^ independent of PTH levels [3]. The 60% of Ca^2+^ in the ECF is not protein-bound, and is thus freely filtered by the renal glomeruli [16]. In the urinary collecting system, up to 98% filtered Ca^2+^ will be reabsorbed via various transepithelial transport routes. As a result, urinary Ca^2+^ excretion will be less than 2% of the daily filtered load of Ca^2+^. There are two major types of transepithelial Ca^2+^ transport pathways—paracellular and transcellular [15]. Paracellular transport is mainly driven by a positive transepithelial potential difference and is controlled by tight junction proteins while transcellular transport pathway mainly occurs in the segment of the distal convoluted tubule (DCT) and consists of apical entry by transient receptor potential cation channel subfamily V member 5 (TRPV5) and basolateral shuttling by a sodium–calcium exchanger and Ca^2+^ ATPase. More details are provided in the subsequent section.

Before exploring the renal handling of Ca^2+^, a general concept of Ca^2+^ homeostasis must first be established first. Increased serum Ca^2+^ can activate the CaSR, which is responsible for a physiological brake on PTH secretion. The decline in PTH results in decreased reabsorption of Ca^2+^ in the kidney, Ca^2+^ absorption in the intestine, and Ca^2+^ resorption from bone. The CaSR is expressed throughout the entire renal tubule system, and elevated extracellular Ca^2+^ can also stimulate renal CaSR, independent of the PTH effects. The activation of renal CaSR in different segments could result in different effects on Ca^2+^ reabsorption [3]. In the different renal segments, the CaSR can be located apically, basolaterally, or at both on cellular levels. The renal distribution of the CaSR is similar across humans, rats, and mouse species, despite a marked variation in its overall expression. The highest expression of the CaSR in kidney is observed in the thick ascending limb of the loop of Henle (TAL) and the lowest expression is found in the glomerulus [17]. However, Ca^2+^ signaling is more distinctive in the glomerulus. Among the other tubular systems, the CaSR is accepted to possess the following functions—1. Ca^2+^ and phosphate (Pi) homeostasis; 2. Cation transport; 3. Fluid/NaCl reabsorption; and 4. Urinary acidification. As the individual functions are dependent on the different segments of the tubular system, we opted to discuss the different segments individually to shed light on their functions.

### 2.1. Glomerulus

The mature glomerulus contains four cell types—podocytes, endothelial cells, epithelial cells, and mesangial cells. Besides other parts of the tubular system, the expression of the CaSR is found to be relatively low in all cell types of the glomerulus [17]. Despite this low expression, the CaSR in the glomerulus plays an important role in extracellular Ca^2+^ entry [18,19]. The modulation of calcium influx in mesangial cells and podocytes is closely related to glomerular function. Under healthy conditions, intracellular Ca^2+^ is maintained at a low concentration; however, under a disease status, its concentration is increased. A continuous increase in intracellular calcium has harmful effects on glomerular cells as it triggers an acute reorganization of the actin cytoskeleton, resulting in glomerulosclerosis and nephropathy [20]. In the segment of the glomerulus, the transient receptor potential canonical (TRPC) channels, TRPC1 to TRPC7, are mainly responsible for Ca^2+^ handling [21]. Among these seven TRPC members, TRPC5 and TRPC6 are the vigorous mediators in the regulation of Ca^2+^ influx in glomerular cells. In fact, gain-of-function mutations in TRPC6 can cause focal segmental glomerulosclerosis (FSGS) [22]. However, under normal physiological conditions, TRPC6 remains silent until a pathologic stimulus triggers its activation. TRPC6 activation through the G protein-coupled receptor (GPCR) signaling mechanism, including angiotensin II (Ang II), adenosine triphosphate (ATP), and protease thrombin, can induce podocyte dysfunction and glomerular filtration barrier breakdown, which presents with deteriorating albuminuria [23]. However, activation of the CaSR in the glomerulus might protect against the glomerular injury mentioned above [18,24,25,26]. Activating the CaSR with a high extracellular Ca^2+^ or calcimimetics (CaSR agonist) can induce an increase in intracellular Ca^2+^ in a dose-dependent manner, and the activation exerts a stable effect on the actin cytoskeleton and attenuates glomerular damage via an unclear mechanism. A recent study has revealed that the renoprotective effect of the CaSR could originate from AMP-activated protein kinase (AMPK) activation [26]; however, this needs to be further evaluated.

### 2.2. Proximal Tubule (PT)

The CaSR is expressed apically in the PT and is responsible for the reabsorption of 60–70% of the Ca^2+^ filtered by the glomerulus [27]. The regulation of phosphaturia is the major function of this segment. Cation homeostasis in the PT is largely modulated by PTH. PTH activates parathyroid hormone receptor 1 (PTH1R), which is in both the apical and basolateral membranes, and inhibits the Na^+^-H^+^ cotransporter (SLC9A3) and Na^+^-Pi cotransporter (SLC34A3). This inhibition leads to an increase in Pi excretion. However, when the CaSR is activated, the PTH effect will be blunt, causing decreased phosphaturia and increased urine acidification.

In the PT, the regulation of serum 1,25(OH)_2_D_3_, which is the physiologically active form of vitamin D, is another major function associated with renal CaSR. The inactive form of vitamin D, that is 25-hydroxyvitamin D (25(OH)D), is converted to 1,25(OH)_2_D_3_ via 1α-hydroxylase (CYP27B1). Of note, the CYP27B1 is only located in the PT. Although PTH could activate CYP27B1, the activation of the CaSR could inversely inhibit CYP27B1. Thus, renal CaSR in the PT will reduce the conversion of 1,25(OH)_2_D_3_. The major function of 1,25(OH)_2_D_3_ is to promote intestinal Ca^2+^ absorption and bone Ca^2+^ resorption [28]. The inhibitory effect of CaSR in the PT could reduce ECF Ca^2+^ by indirectly affecting organs besides the kidneys.

Ultimately, Ca^2+^ is mainly reabsorbed through the paracellular transepithelial routes in the PT. Activation of the CaSR leads to Na^+^ reabsorption and proton excretion. Increased intracellular Na^+^ causes increased osmolarity and provides a paracellular driving force to strengthen Ca^2+^ and H_2_O reabsorption.

### 2.3. Thick Ascending Limb of the Loop of Henle (TAL)

After the PT, the thin descending, and ascending limbs of the loop of Henle all have limited Ca^2+^ permeability [27]. In both animal and the human studies, the TAL was found to have the highest level of CaSR expression, with its expression mainly found on the basolateral membrane. Approximately 20–30% of filtered Ca^2+^ is reabsorbed in the TAL through the paracellular transepithelial routes [29]. To understand the mechanism of Ca^2+^ reabsorption in TAL, the role of Na^+^ and K^+^ should be considered. There are two important transporters on the apical membrane of TAL—renal-outer-medullary-potassium (ROMK) and Na^+^-K^+^-Cl^−^ cotransporters 2 (NKCC2) [27]. ROMK is responsible for the recycling of K^+^ into urine and is the rate-limiting step of NKCC2 (encoded by the SLC12A1 gene). NKCC2 mediates the transepithelial NaCl reabsorption in the TAL and plays an essential role in the urinary concentration. These two transporters are activated by intracellular cAMP through the GPCR pathway. The GPCR pathway can be activated by PTH, calcitonin, and arginine vasopressin (AVP). Thus, PTH can lead to a voltage difference via K^+^ recycling and promote Ca^2+^ reabsorption paracellularly. In contrast, CaSR activation can concurrently inhibit cAMP and ROMK. This effect gives rise to a reduction in luminal transepithelial potential difference and decreases the paracellular driving force of Ca^2+^. As a result, typical clinical manifestations of Bartter syndrome, including hypokalemia, hypochloridemia and metabolic alkalosis, will develop [30].

Paracellular Ca^2+^ flow is strictly regulated by claudins (CLDNs), including CLDN14, CLDN16, and CLDN19. CLDNs are tight-junction proteins consisting of a family of 27 members that function as ion channels to allow selective permeation. CLDN16 and CLDN19 are fully expressed in the TAL and form a heteromeric cation channel. The functions of these cation channels include—(1) facilitating the permeation of Ca^2+^ and Mg^2+^; and (2) creating a lumen-positive voltage potential in the late TAL and inducing the electrical driving force for Ca^2+^ and Mg^2+^ reabsorption [31]. However, compared to CLDN16 and CLDN19, CLDN14 plays an opposite role in the control of Ca^2+^ influx. CLDN14 is exclusively expressed in the TAL and is undetectable at baseline. However, hypercalcemia activates the CaSR expressed in the basolateral membrane and increases CLDN14 expression [32]. According to recent studies, CLDN14 is primarily expressed in cells that also express CLDN16, and directly interacts with CLDN16 but not CLDN19. CLDN14 overexpression leads to the blockage of paracellular Ca^2+^ flux, which promotes urinary Ca^2+^ excretion. Based on accumulating evidence, a dysfunction in CLDN14 is highly associated with kidney stone formation [33].

### 2.4. Distal Convoluted Tubule (DCT)

In the DCT, approximately 10% of filtered Ca^2+^ is reabsorbed through the transcellular transepithelial routes. The CaSR is expressed both on the apical membrane and on the basolateral membrane, and its functions vary between the two locations. First, after activation of the apical CaSR, Ca^2+^ uptake occurs via the TRPV5 transporter. TRPV5 can also be activated by increased cAMP, which is caused by PTH activation. Second, Ca^2+^ will be bound to calbindin-d28k (CALB1) and transported from the apical side to the basolateral membrane. Third, there are two Ca^2+^ transport systems located on the basolateral membrane—plasma membrane calcium ATPase 4 (PMCA4) and the calcium/sodium exchanger (NCX1). Recent evidence has revealed that NCX1, not PMCA4, is the main regulating transporter [34]. The activity of NCX1 is highly dependent on intracellular Na^+^ level, which is closely associated with apical SLC12A3 (NCC) and the basolateral Na^+^/K^+^ ATPase. The former is responsible for NaCl reabsorption while the latter pumps sodium into the basolateral space. The activation of the basolateral CaSR will inhibit the KCNJ10 and PMCA4 transporters. KCNJ10 and PMCA4 allow K^+^ and Ca^2+^ to flow into the basolateral space, respectively. The activation of CaSR will thus directly promote Ca^2+^ reabsorption and indirectly inhibit NaCl reabsorption.

### 2.5. Collecting Duct

The collecting duct is the final part of the renal tubular system. This structure generally allows water reabsorption and proton excretion [35]. The entire collecting duct is comprised of two different types of cells—principal cells and intercalated cells (ICs). Principal cells are thought to have a critical role in Na^+^ and water regulation while the ICs mainly regulate acid–base homeostasis. In the principal cells, the activation of CaSR expressed apically inhibits AQP2, which is responsible for water reabsorption from the urine. The mechanism of AQP2 begins with vasopressin bound to the type 2 vasopressin receptor (V2R), located on the basolateral membrane of the principal cells. After vasopressin binds to AVP, the cAMP/PKA signal is activated, causing the AQP2-containing vesicles to fuse with the plasma membrane. AQP2 is then apically inserted into the cell. As a result, there is an increase in water permeability, which allows water reabsorption from urine. Ultimately, the activation of the CaSR reduces the water reabsorption and promotes urinary dilution.

Three types of ICs are found in the collecting duct—type A (A-ICs), type B (B-ICs), and the non-A, non-B ICs [36]. The A-ICs mainly function in urinary acidification whereas B-ICs and non-A, non-B ICs function in urinary HCO_3_^−^ secretion. ICs can excrete H^+^ into urine via the H^+^-ATPase on the apical membrane and the activation of CaSR can promote the activity of the H^+^-ATPase and induce urinary acidification.

### 2.6. Brief Conclusion

Following CaSR activation, PTH-mediated Pi excretion in the PT will be counteracted, leading to reduced phosphaturia. The activation of CaSR in the TAL will first reduce the transport of NaCl/Ca^2+^/Mg^2+^ by inhibiting NKCC2 and ROMK, and then reduce Ca^2+^ transport via the paracellular routes by inhibiting CLDN16. These effects are related to the reduced reabsorption of Ca^2+^. In the DCT, the apical CaSR increases Ca^2+^ reabsorption via TRPV5 independent of NaCl reabsorption. Finally, in the collecting duct, the activation of CaSR will cause urinary dilution and induce urinary acidification through the promotion of the H^+^-ATPase (Figure 1).

## 3. Ca^2+^ Signaling in Vascular Calcification

### 3.1. Classification of Vascular Calcification

Vascular calcification is associated with cardiovascular morbidity and mortality [37]. Various physiological and pathological conditions will lead to vascular calcification, including aging, DM, and CKD. Vascular calcification can be categorized into two distinct forms, intimal calcification and medial calcification. Intimal calcification is usually associated with atherosclerotic plaques, which could cause vascular obstruction, and ultimately rupture [38]. On the other hand, medial calcification is more common in the patients with CKD and DM. The mechanism of vascular calcification is complex and dependent on the specific form and etiology. Generally, intimal calcification is similar to the multistep process of bone formation and is accelerated by inflammation [39]. The process begins with microcalcifications (0.5–50 μm) in the atheroma, which are derived from the production of apoptotic bodies and matrix vesicles by vascular smooth muscle cells (VSMCs) and macrophages under inflammatory stimulation [40]. These microcalcifications are believed to be vulnerable and unstable. During the process of repair, macrophage polarization facilitates plague macrocalcification by osteoblastic differentiation [41]. Medial calcification is frequently associated with the osteo-/chondrogenic transdifferentiation of VSMCs [42]. This transdifferentiation can be activated by hyperphosphatemia, hypercalcemia, hyperglycemia, and inflammation [43], and involves with various osteogenic transcription factors, including core binding factor alpha-1(CBFA1), msh homeobox 2 (MSX2), and osterix (Osx). Further, this process is associated with the dysfunction of calcification inhibitors such as pyrophosphate (PPi), matrix Gla protein (MGP), and fetuin-A [44]. Of note, a discussion of the mechanism of vascular calcification is outside the scope of this review. Herein, we opted to focus on the Ca^2+^ signaling and the role of CaSR in vascular calcification.

### 3.2. The Role of CaSR in Vascular Calcification

In the vascular system, CaSR is expressed on vascular endothelial cells (VECs), VSMCs, and peri-vascular nerves to regulate vessel elasticity. The activation of the CaSR in these cells mainly results in vascular relaxation under complex pathways, which is closely related to blood pressure control. However, in vascular calcification, the CaSR in mainly involved in the dysfunction of Ca^2+^ homeostasis. Previous studies have found that calcium channel blockers can block Ca^2+^ entry into VSMCs and reduce vascular calcification [45,46]. The Ca^2+^ efflux related proteins, including NCKX4, NCX1, and PMCA1, were found to be decreased in the aortic media of a vascular calcification mouse model and cultured VSMCs exposed to high concentrations of Pi and Ca^2+^ [47]. These findings suggest that an imbalance in Ca^2+^ influx into VSMCs could cause vascular calcification. However, the VSMC CaSR is believed to prevent the development of vascular calcification [35,37]. This concept originated from the studies of VSMCs, with decreased CaSR expression in CKD patients leading to increased vascular calcification [48]. This finding is also supported by many recent studies that used calcimimetics to increase CaSR expression and reduce vascular calcification in VSMCs [49,50]. The protective function of calcimimetics on vascular calcification has been repeatedly proven in numerous CKD cohort studies [51,52]. The MGP is generally accepted to be responsible for the protective effects of CaSR [53]. MGP is one of the most important inhibitors against vascular calcification [54] and is a vitamin K-dependent protein that is mainly synthesized by VSMCs and chondrocytes. MGP-knockout mice were found to die in two months owing to extensive arterial calcification and aortic rupture [55]. The activation of CaSR in VSMCs directly leads to an increase in the synthesis of MGP [53].

## 4. Ca^2+^ Signaling in Nephrolithiasis

Nephrolithiasis is a major urological disease with a lifetime prevalence of approximately 10% [56]. Calcium-containing stones are always the most common urinary tract stones [57]. Calcium oxalate (CaOx) is the most common stone type, ranging from 65–85%, followed by CaP [58]. Renal Ca^2+^ signaling dysregulation is always considered to be the most critical factor for the induction of nephrolithiasis formation. Although the mechanism of stone formation has been studied for years, it remains incompletely understood. Kidney stone formation is a process involving physicochemical changes and supersaturation of urine. The physicochemical changes imply that an imbalance exists between urinary promoters (e.g., Ca^2+^, Pi, uric acid, oxalate) and urinary inhibitors (e.g., citrate, osteopontin (OPN)). Hypercalciuria was found to be the most common metabolic risk factor in stone formers [59]. Of note, a solution is considered to be supersaturated when it contains more material than the solvent can dissolve under normal circumstances. Under the supersaturation state, solute precipitation in urine leads to nucleation followed by crystallization. Crystallization occurs when the concentration of two ions exceeds their saturation point [60]. Crystal aggregation is involved in crystal retention and leads to crystal–cell interactions. This aggregation in crystals damages renal tubule cells and results in nephrolithiasis.

With advance in genome sequencing technologies in recent years, studies have emphasized the importance of genetic causes of kidney stone disease. Past literature has reported that a monogenic cause could be detected in 10–15% of individuals with adult-onset nephrolithiasis and in 15–20% of individuals <18 years old [61,62,63,64]. However, the genetic influence on idiopathic calcium stone formers might be underestimated according to the results from twin studies [65]. Most genetic disorders related to nephrolithiasis are associated with hypercalciuria [61]. Based on the knowledge regarding the renal Ca^2+^ handling pathway, it is easier to understand the genetic influences underlying nephrolithiasis formation. First, the CaSR has the most important and complex role in renal Ca^2+^ handling. In fact, the activation of the CaSR can result in reduced calcium reabsorption in the TAL and increased calcium reabsorption in the DCT. Approximately 200 nonsynonymous single-nucleotide polymorphisms have been found in the human CaSR gene, and some of these are associated with idiopathic calcium kidney stones [66]. Decreased activity of CaSR gene expression prompts calcium nephrolithiasis formation [67]. The inability to reabsorb urinary Ca^2+^ in the TAL and to dilute urine may thus be the causes; however, the exact mechanism is still unclear. Interestingly, the protective role of the CaSR in calcium nephrolithiasis is akin to its role in vascular calcification. Second, the mutation of genes associated with calcium homeostasis in the TAL results in a syndrome (Bartter syndrome) that presents with hypokalemic alkalosis, hyperreninemic hyperaldosteronism, and hypercalciuria [30]. These mutations affect the activities of SLC12A1 (Type I Bartter syndrome), ROMK (Type II), and CaSR (Type V). All types of Bartter syndrome mentioned above could result in hypercalciuria and an increased risk of nephrolithiasis. Finally, renal CLDNs are closely related to hypercalciuria and nephrolithiasis [68]. In addition, CLDN14 expression is responsible for decreased urinary Ca^2+^ reabsorption during elevated serum Ca^2+^ levels, and dysregulation of CLDN14 contributes to the development of nephrolithiasis [69]. CLDN16 and CLDN19 form a complex for the paracellular reabsorption of Ca^2+^. Patients with mutations in the CLDN16 or CLDN19 genes, which are characterized by excessive renal magnesium and calcium excretion, are diagnosed with a syndrome of familial hypomagnesaemia with hypercalciuria and nephrocalcinosis (FHHNC) [70]. Recent novel studies revealed CLDN2 is the mediator of calcium reabsorption in the PT and plays a critical role in idiopathic hypercalciuria and nephrolithiasis [71]. CLDN2-knockout mice were found to have hypercalciuria attributable to both a renal leak of calcium and increased net intestinal calcium absorption. However, further investigations are still warranted.

## 5. The Connection between Nephrolithiasis and Vascular Calcification

In recent decades, growing evidence has linked nephrolithiasis to various metabolic diseases, including DM, obesity, cardiovascular disease, CKD, and metabolic syndrome [72,73,74]. All these systemic diseases are closely related to vascular calcification. As a result, observations of increased vascular calcification in stone formers are of marked interest, despite the unclear underlying mechanism. First, previous animal studies have revealed osteopontin (OPN), which is abundantly expressed in calcified vessels, knockout mice frequently developed vascular calcification and also had increased risk of renal crystal deposition [75,76,77]. High-phosphate-fed OPN-knockout mice had more severe vascular calcification and renal calcification compared with OPN-null mice, which indicated OPN may simultaneously protect against nephrocalcinosis and vascular calcification [78]. MGP, the most vigorous inhibitor of tissue calcification, participated in both vascular calcification and kidney stone formation, proven by numerous animal studies [55,79]. Otsuka Long-Evans Tokushima fatty (OLETF) rats, which are the model of metabolic syndrome, formed more renal crystal deposits and also exhibited a greater coronary vascular stenosis. All these animal studies have hinted a link between these two distinct diseases [80,81]. Second, prior human studies, which were assessed herein to identify the association between nephrolithiasis and vascular calcification, were mainly based on large-scale epidemiologic results. The Coronary Artery Risk Development in Young Adults (CARDIA) study was the pioneer study for dissecting this association [82]. By employing a 20-year follow-up retrospective cohort, the CARDIA study revealed that a young kidney stone former has a higher prevalence of subclinical carotid atherosclerosis than non-stone formers (odds ratio (OR), 1.67; 95% confidence interval (CI), 1.17–2.36). After this study, two different groups found that patients with nephrolithiasis had increased arterial stiffness and vascular calcification, which were assessed by pulse-wave velocity and abdominal aortic calcification, respectively [10,83]. The Multi-Ethnic Study of Atherosclerosis (MESA) study, which recruited more than 3000 participants, further confirmed that recurrent kidney stone formers had a higher risks of coronary artery calcification than non-stone formers (OR, 1.80; 95% CI, 1.22–2.67) [84]. With the help of computed tomography, several studies have proven the higher prevalence of abdominal aortic calcification in kidney stone formers than non-stone formers [85,86]. Recently, our group had found an increase in carotid intima–media thickness, independent of dyslipidemia, in calcium kidney stone formers compared to stone-free controls [87]. Altogether, considerable evidence supports that patients with nephrolithiasis are at a higher risk of vascular calcification.

Although a consistent association between nephrolithiasis and vascular calcification has been established, the underlying mechanism remains uncertain. Nonetheless, several theories can be considered (Figure 2). First, both lifestyle and dietary habits can lead to these two distinct diseases. High-fat diets can cause vascular calcification and induce urinary acidification to promote calcium kidney stone formation [88]. In addition, obesity and metabolic syndrome are well-known risk factors of nephrolithiasis and vascular disease [13]. Second, macrophage polarization and phagocytosis can lead to the development of kidney stone and vascular calcification. Human monocytes can be activated by the CaOx kidney stone and produce inflammatory cytokines locally. Macrophage M1 polarization can facilitate kidney stone formation [89] and promote the differentiation of VSMCs into osteoblastic phenotypes via the Janus kinase 3-signal transducer and activator of transcription protein 3 (JAK3-STAT3) pathway [90]. Third, matrix vesicles are present at the sites of vascular calcification and the kidney papillary area [91,92]. Matrix vesicles are the extracellular membrane-bound vesicles secreted by many cell types and participate in the progression of cardiovascular calcification. For nephrolithiasis, crystal deposition in the renal papilla is believed to begin with matrix vesicles that could cause nucleation and crystallization; the size of the area then grows with additional crystals deposited at the periphery in a collagen framework [93]. Fourth, decreased expression of calcification inhibitors, including OPN, MGP, and fetuin-A, has been observed in vascular calcification and nephrolithiasis. OPN is a critical inhibitor of kidney stone formation; however, this inhibitor can also ameliorate vascular calcification by preventing CaP crystal growth [76,77]. No matter whether it is in vascular calcification or nephrolithiasis patients, circulating levels of OPN are reduced [78,94]. Fetuin-A is also a well-known calcification inhibitor. According to previous literature, vascular calcification and arterial stiffness are related to low serum levels of Fetuin-A [77]. Patients with kidney stones have been found to have significantly lower serum and urinary levels of Fetuin-A than non-stone formers [95]. Compared to OPN and Fetuin-A, MGP might be the most important inhibitor of vascular calcification and nephrolithiasis [77]. MGP-deficient mice can spontaneously develop arterial calcification [96]. Similarly, numerous studies have found that patients with an MGP gene polymorphism are more likely to have kidney stone disease [97]. All of this evidence indicates a decrease in calcification inhibitors may be the link between vascular calcification and nephrolithiasis. Finally, a dysfunction in CaSR may be the possible mechanism underlying vascular calcification and nephrolithiasis as discussed above. In brief, the CaSR impairment can lead to a dysregulation in renal calcium handling and the failure to prevent vascular calcification. However, the pathophysiology of vascular calcification and kidney stone formation is not fully understood. The hypotheses mentioned above must be verified in future studies.

## 6. Conclusions and Perspectives

Owing to more advanced animal and molecular studies, the process of renal Ca^2+^ signaling has become clearer over the years. In addition, the pathogenic link between vascular calcification and nephrolithiasis has been further corroborated by a growing body of evidence. As calcium is the main component in these two distinct conditions, a dysfunction in calcium homeostasis might be the mechanism connecting the two. Although calcium dysregulation cannot be used to explain the mechanisms in all patients, knowledge of the Ca^2+^ signaling can aid in the discovery of the optimal drug targets. Currently, vascular calcification or nephrolithiasis occurs due to a lack of effective medications. Consequently, therapeutic strategies targeting the CaSR might reduce the progression of vascular calcification and nephrolithiasis.

## Figures and Tables

**Figure 1 ijms-22-03590-f001:**
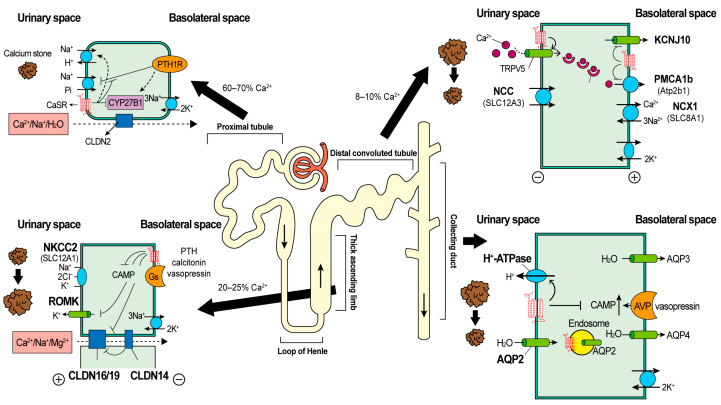
After glomerulus filtration, urinary Ca^2+^ flows through the proximal tubule (PT), with 60–70% reabsorption occurring via the paracellular routes. The calcium-sensing receptor (CaSR) is mainly expressed on the basolateral membrane of PT. The major role of the CaSR in the PT is to counteract parathyroid hormone (PTH)-mediated Pi excretion, leading to reduced phosphaturia. Claudin 2 (CLDN2) is a novel paracellular cation channel, and that it is permeable to Ca^2+^ and Na^+^. After the PT, urine passes to the thick ascending limb of the loop of Henle (TAL), which has the highest expression of CaSR and causes 20–25% Ca^2+^ reabsorption via the paracellular routes. When hypercalcemia activates CaSR in the TAL, it leads to the inhibition of renal-outer-medullary-potassium (ROMK), which prevents apical K^+^ recycling—the rate-limiting step for Na^+^-K^+^-Cl^−^ cotransporter 2 (NKCC2) activity and reduces intracellular cyclic AMP (cAMP) levels. This effect leads to a loss of the driving force for paracellular Ca^2+^ reabsorption. The activation of CaSR can also enhance CLDN14, which is responsible for controlling Ca^2+^ influx. Through the segment of the distal convoluted tubule (DCT), 8–10% Ca^2+^ will be reabsorbed via the transcellular routes. Transient receptor potential cation channel subfamily V member 5 (TRPV5), activated by CaSR, is the key channel to reuptake Ca^2+^. Plasma membrane calcium ATPase 1b (PMCA1b) and the calcium/sodium exchanger (NCX1) are the Ca^2+^ channels located basolaterally and are mediated by CaSR. Finally, in the collecting duct, apical CaSR activation reduces vasopressin-mediated apical insertion of the aquaporin 2 (AQP2) water channel and decreases the water reabsorption. In addition, CaSR activation could also inhibit H^+^-ATPase and lead to urinary acidification. PT, proximal tubule; CaSR, calcium-sensing receptor; PTH, parathyroid hormone; Pi, phosphate; CLDN, claudin; TAL, thick ascending limb of the loop of Henle; DCT, distal convoluted tubule.

**Figure 2 ijms-22-03590-f002:**
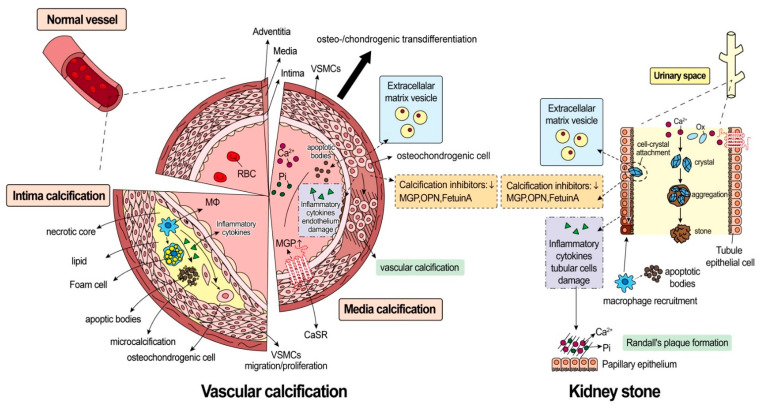
Crosstalk between vascular calcification and nephrolithiasis. The process of vascular calcification is demonstrated on the right while that of kidney stone formation is depicted on the left. The molecular characteristics shared by these two distinct diseases are highlighted by the same-colored block. **Vascular calcification** can be categorized into intimal calcification and media calcification. 1. Intimal calcification is frequently associated with atherosclerosis. Macrophage (Mφ) digests lipoproteins and converts them to cholesterol-rich foam cells. Initially, foam cells can be phagocyted by adjacent vascular smooth muscle cells (VSMCs) and release apoptotic bodies. In addition to an increase in the number of apoptotic bodies, there is an accumulation of apoptotic body debris, which results in the formation of a necrotic core. Vesicles in the necrotic core release microcalcification of calcium phosphate as nucleation nidus. 2. Medial calcification is largely related to osteo-/chondrogenic transdifferentiation, which indicates the change in phenotype of VSMCs into osteo-/chondroblasts like cells. Transdifferentiation can be induced by high phosphate (Pi) or calcium (Ca^2+^) levels. The osteo-/chondroblast-like cells actively promote media calcification by reduced activities of calcification inhibitors (e.g., matrix Gla protein (MGP), osteopontin (OPN), and fetuin A), apoptotic body release, calcifying extracellular matrix vesicle release, and inflammatory cytokine release. The activation of CaSR on VSMCs can stimulate MGP release, which can ameliorate vascular calcification. The mechanism of **kidney stone** formation remains largely unknown. However, the supersaturated urinary stone promoters, e.g., Ca^2+^ and oxalate (Ox), can gradually form crystals, and aggregated crystals could interact with tubule epithelial cells and cause epithelial damage. The cell-crystal interaction could cause calcifying extracellular matrix vesicles release and inflammatory cytokines release, with the assistance of reduced activities of calcification inhibitors (e.g., MGP, OPN, and fetuin A). Mφ recruitment and polarization are also found to occur in the crystal-attached areas. Finally, Ca^2+^ and Pi combine to form hydroxyapatite, which is deposited on the renal papilla and referred to as Randall’s plaque.

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
