# Peer review of "Crosstalk between Renal and Vascular Calcium Signaling: The Link between Nephrolithiasis and Vascular Calcification"

_ijms, 2021, doi:10.3390/ijms22073590_

Round 1

Reviewer 1 Report

This manuscript addressed the Crosstalk between renal and vascular calcium signaling: the link between neprolithiasis and vascular calcification. This review summarizes renal calcium handling and calcium signaling in vascular system, with a particular highlight in the link between nephrolithiasis and vascular calcification.

The demonstrated that Calcium is an important mediator of multiple cellular homeostasis and involved in a variety of diseases.

Aberrant Ca2+ sensing by the kidney and VSMCs, beacuse of altered CaSR expression of function is associated with the formations of nephrolithiasis and vascular calcification.

Medial calcification is frequently associated with the osteo/chondrogenic transdifferentiation of VSMCs. The transdifferentiation can be activated by hyperphosphatemia, hypercalcemia, hyperglycemia, inflammation and hypoxia (Balogh et al, Arteriosclerosis, Thrombosis, and Vascular Biology.2019; 39:1088-1099). This work is highly relevant to this chapter I would suggest including in the reference list.

Comments:

  1. Do not mention „in other word”, or do not use so frequently within a chapter. Please find and use another term.
  2. Ca2+ signaling in vascular calcification chapter is so complex. I would suggest to divide it into further schapters and Ca2+ signaling in nephrolithiasis.
  3. I would suggest to mention some animal model in connection with this topic. If it is possible to include in this review.
  4. There are some typos throughout the text.
  5. I would suggest a more detailed discussion.

Author Response

The content is too complicated. Please see the attachment. Thanks!

Reviewer 2 Report

The review analyzes in depth the links between nephrolithiasis and atherosclerosis with particular regard to the role of the calcium-sensing receptor.
The article is well-written and complete. The figures are also well done.
I just report some typos (for example on page 6 line 260: acerated).

Reviewer 3 Report

Liu CJ and co-workers wrote a comprehensive review on the complex interplay linking nephrolithiasis and vascular calcification. The paper is very rich of notions and I consider that such a work can be of highly interest for the readers of IJCM. Nevertheless, I have some concerns on this paper.

First of all, the Authors should clearly state whether this is a narrative or a systematic review. In both cases they must add a short section on the methodology for retrieving the articles cited in the paper.

Second, I think that the whole paper is somehow too ponderous. Although I’ve really appreciated Author’s effort in giving an accurate description of the mechanisms underlying renal and vascular calcium signaling, a “leaner” version of the text could improve readability for many readers.

Third, the figures are very well done, but I think that they are too small and a wider version would add a great value to the paper. Moreover, colored figures are highly desirable, particularly in such a comprehensive review.

Fourth a mother-tongue revision, together with some scientific changes, are advisable; I’m giving some examples in the non-exhaustive list below (possible corrections are in capital letters):

Line 39-40: Calcitonin must be added among calcium regulators.

Line 42: “CaSR can specifically detect a slight increase in serum Ca2+ and IT CAN be activated by Ca2+ with low affinity.”

Lines 49-50: This sentence is too assertive. A more reasonably conservative sentence should be: “Dysregulation of Ca2+ MAY also CONTRIBUTE to THE PATHOGENESIS numerous diseases, e.g., malignancy, diabetes mellitus (DM), and hypertension (HTN).

Lines 58-59: “The most common composition of RENAL STONES…”

Line 60: “Randall’s plague” is definitely RANDALL’S PLAQUE and it should be corrected throughout the text.

Lines 67-70: “To consider in the pivot role of CaSR in regulating calcifications in various organs, the dysregulation of calcium signaling in the vessel and the kidney may be the possible link connecting vascular calcification and nephrolithiasis.”. THIS IS A GOOD EXAMPLE OF A HAZY SENTENCE THAT NEEDS TO BE REPHRASED.

Lines 97-98: “In the different renal segments, CaSR can locate apically, basolaterally, or AT both cellular levelS.”

Line 99: “though the overall expression VARIES markedly.”

Line 113: “In healthy condition, the intracellular Ca2+ IS MAINTAINED…”

Line 113: “which characterized with deteriorating albuminuria…” I think that CHARACTERIZED is a wrong verbal form.

Line 164: “(SLC12A1)”. It seems that this acronym has never been mentioned before, therefore it is unclear.

Lines 170-171: “This effect gives rise to a loss of driving force for paracellular driving force.”. This sentence should be rephrased in a clearer way.

Line 164: “compromising”. Is it the right verb here? I’m not sure.

This list could go further and further, but I think I’ve given a sufficient idea of what I meant with “mother-tongue revision, together with some scientific changes”.

Author Response

Point 1: First of all, the Authors should clearly state whether this is a narrative or a systematic review. In both cases they must add a short section on the methodology for retrieving the articles cited in the paper.

Response 1:

Thank you for your friendly reminder. Our article is a narrative review, not systemic review. Initially, we drafted this review according to the reviews, that were already published on the International Journal of Molecular Sciences, so we didn’t state a clear methodology. It is our mistake. We have refined as the follows.

Point 2: I think that the whole paper is somehow too ponderous. Although I’ve really appreciated Author’s effort in giving an accurate description of the mechanisms underlying renal and vascular calcium signaling, a “leaner” version of the text could improve readability for many readers.

Response 2:

Thank you for your kind reminder. You have raised an important problem that too complicated content would make the readers hard to realize. We have sent for a formal English editing and requested to have at least 10% reduction in article volume. Please see the attached certification. The revised part of manuscript was as the attachment.

Point 3: the figures are very well done, but I think that they are too small and a wider version would add a great value to the paper. Moreover, colored figures are highly desirable, particularly in such a comprehensive review.

Response 3:

Thank you. I appreciate your compliment. We have revised our original figure into colored figure. Besides, we provide a large and high-quality figure to you and all readers.

Point 4: a mother-tongue revision, together with some scientific changes, are advisable; I’m giving some examples in the non-exhaustive list below (possible corrections are in capital letters):

Response 4:

Thank you for your friendly reminder. We have sent for official English editing . The revised version is modified by mother-tongue English profession editor. Please see the attached certification. The point-to-point revision is listed as follows.

Please, see the attachment, thank you. 

Round 2

Reviewer 3 Report

This reviewer appreciates Authors’ effort in answering his major concerns. I have particularly appreciated the colored version of the figures. They are now very helpful. The paper is now more balanced and clearer.